# Triggering of lymphocytes by CD28, 4-1BB, and PD-1 checkpoints to enhance the immune response capacities

Elina Kaviani[1,2], Ahmad Hosseini[2], Elham Mahmoudi Maymand[2], Mani Ramzi[3], Abbas Ghaderi[2], Amin Ramezani[1,2]*

1 Department of Medical Biotechnology, School of Advanced Medical Sciences and Technologies, Shiraz University of Medical Sciences, Shiraz, Iran, 2 Shiraz Institute for Cancer Research, School of Medicine, Shiraz University of Medical Sciences, Shiraz, Iran, 3 Hematology Research Center, Shiraz University of Medical Sciences, Shiraz, Iran

* aramezani@sums.ac.ir

**Data Availability Statement:** All relevant data are within the paper and its Supporting Information files.

## Abstract

Tumor infiltrating lymphocytes (TILs) usually become exhausted and dysfunctional owing to chronic contact with tumor cells and overexpression of multiple inhibitor receptors. Activation of TILs by targeting the inhibitory and stimulatory checkpoints has emerged as one of the most promising immunotherapy prospectively. We investigated whether triggering of CD28, 4-1BB, and PD-1 checkpoints simultaneously or alone could enhance the immune response capacity of lymphocytes. In this regard, anti-PD-1, CD80-Fc, and 4-1BBL-Fc proteins were designed and produced in CHO-K1 cells as an expression host. Following confirmation of the Fc fusion proteins' ability to bind to native targets expressed on engineered CHO-K1 cells (CHO-K1/hPD-1, CHO-K1/hCD28, CHO-K1/hCTLA4, and CHO-K1/h4-1BB), the effects of each protein, on its own and in various combinations, were assessed in vitro on T cell proliferation, cytotoxicity, and cytokines secretion using the Mixed lymphocyte reaction (MLR) assay, 7-AAD/CFSE cell-mediated cytotoxicity assay, and a LEGENDplex™ Human Th Cytokine Panel, respectively. MLR results demonstrated that T cell proliferation in the presence of the combinations of anti-PD-1/CD80-Fc, CD80-Fc/4-1BBL-Fc, and anti-PD-1/CD80-Fc/4-1BBL-Fc proteins was significantly higher than in the untreated condition (1.83-, 1.91-, and 2.02-fold respectively). Furthermore, anti-PD-1 (17%), 4-1BBL-Fc (19.2%), anti-PD-1/CD80-Fc (18.6%), anti-PD-1/4-1BBL-Fc (21%), CD80-Fc/4-1BBL-Fc (18.5%), and anti-PD-1/CD80-Fc/4-1BBL-Fc (17.3%) significantly enhanced cytotoxicity activity compared to untreated condition (7.8%). However, concerning the cytokine production, CD80-Fc and 4-1BBL-Fc alone or in combination significantly increased the secretion of IFN-γ, TNF-α, and IL-2 compared with the untreated conditions. In conclusion, this research establishes that the various combinations of produced anti-PD-1, CD80-Fc, and 4-1BBL-Fc proteins can noticeably induce the immune response in vitro. Each of these combinations may be effective in killing or destroying cancer cells depending on the type and stage of cancer.

**Funding:** Elina Kaviani, a medical biotechnology student, conducted this research as part of her Ph. D. thesis. Shiraz University of Medical Sciences (Grant No.16179) and the Shiraz Institute for Cancer Research provided funding for this study (Grant No. ICR-100-509). The funders had no role in study design, data collection and analysis, decision to publish, or preparation of the manuscript.

**Competing interests:** The authors have declared that no competing interests exist.

## Introduction

In recent years, through targeting the inhibitory and stimulatory receptors on T cells, immunotherapy made enticing breakthroughs in cancer treatment. This method is based on an overarching concept, namely, that one of the mechanisms of tumor escape from the immune surveillance is to disrupt the expression of inhibitory and stimulatory receptors, which leads to T cells exhaustion. The exhausted T cells overexpress inhibitory receptors including PD-1 and CTLA-4 on their cell surface [1]. The 2018 Nobel Prize in physiology and medicine was awarded to James Allison and Tasuku Honjo for their breakthrough in cancer immunotherapy by targeting CTLA-4 and PD-1, respectively [2].

Checkpoint inhibitors such as nivolumab (anti-PD-1), and ipilimumab (anti-CTLA-4) that prevent PD-1/PD-L1, PD-1/PD-L2, and CTLA4/B7 interactions are now FDA approved for treating a wide range of cancers [3]. Although such antibodies have transformed cancer treatment, only a small percentage of patients respond to them, and different mechanisms from the beginning (initial resistance) or during treatment (acquired resistance) cause restriction to monotherapy with these antibodies [4]. Thus, the synergistic effect of their combination was probed. Treatment of advanced melanoma with nivolumab plus ipilimumab demonstrated an objective response rate (ORR) of up to 57.6%, which is higher than nivolumab (43.7%) and ipilimumab (19%) monotherapy. Median progression-free survival (PFS) for melanoma patients in the three treatment groups, nivolumab plus ipilimumab, nivolumab and ipilimumab, was reported 11.5, 6.9, and 2.9 months, respectively [5].

An alternative therapeutic strategy is to use the natural ligands of the immune checkpoints such as CD137 (4-1BBL ligand) and CD80 (CTLA4 and CD28 ligand). An *in vivo* study has shown that CD80-Fc protein is effective in slowing tumor growth and activating tumor-infiltrating T cells. CD80-Fc induces activation and proliferation of T cells by saturating CTLA-4 receptors and binding to the CD28 stimulatory receptor, in addition to preventing PD-1/PD-L1 interaction by binding to PD-L1 [6].

New cancer therapeutic approaches have been designed to target the co-stimulatory receptors of tumor necrosis factor receptor superfamily (TNFRSF), like 4-1BB. Activation of 4-1BB induces the expression of pro-survival proteins (including BCL-2 protein family) through NF-κB signaling, which in turn increases the survival of $CD8^+$ T cells and antitumor activity [7, 8]. Urelumab and utomilumab are two anti-4-1BB agonistic monoclonal antibodies whose clinical development has been discontinued due to hepatotoxicity and low efficacy [9, 10]. However, the treatment of 46 metastatic melanoma patients with a combination of urelumab and nivolumab in phase I/II trial showed promising results with an ORR of 50% (23/46 patients) [11]. The 4-1BB natural ligand (4-1BBL) is a form of 4-1BB agonist and like utomilumab, binds to the CRD2/3 region of 4-1BB [12] and has the prospects to be used in tandem with a checkpoint blockade.

According to the data presented above, combination therapy by targeting inhibitory and stimulatory checkpoints may be more efficient for cancer immunotherapy. This study aimed to generate three homodimeric Fc-fusion proteins: an anti-PD-1, a CD80 extracellular domain (ECD)-Fc, and a 4-1BBL extracellular domain (ECD)-Fc fusion proteins, and to study their potential for activating co-stimulatory receptors and blocking inhibitory checkpoints in T cells. The next objective is to investigate the in vitro combination of these proteins for immune response enhancement.

## Materials and methods

### Plasmid construction

The anti-PD-1 scFv of nivolumab (variable heavy and light chains of nivolumab were used to design anti-PD1 scFv [DrugBank database, accession number: DB09035], CD80 ECD

[accession number: NP_005182.1], and 4-1BBL ECD [accession number: NP_003802.1] were genetically fused to fragment crystallizable (Fc) region (Hinge-CH2-CH3) of human immunoglobulin gamma 4 (IgG4) with an *S228P* mutation [13], along with human albumin signal peptide at N-terminal domain [14]. These fragments were sequence-optimized by GenScript's Multiparameter Gene Optimization algorithm, OptimumGeneTM (NJ, USA), and were synthetized by Biomatik (Ontario, Canada). The synthesized coding genes were cloned into pCHO1.0 expression vectors at AvrII/BstZ17I insertion site. These constructs were transfected to CHO-K1 cell line by Gene Pulser Xcell electroporation system (Bio-Rad, CA, USA). 48 hours after transfection, the production of anti-PD-1 scFv–Fc, CD80-Fc, and 4-1BBL-Fc proteins in the cell culture supernatant were evaluated by sandwich ELISA (Enzyme-Linked Immunosorbent Assay) and dot blot assay. For providing a stable cell line a two-phase selection strategy based on increasing concentrations of puromycin (Gibco) and MTX (Sigma, MO) was used [15]. In brief, 20 μg/mL Puromycin and 200 nM MTX were used in selection phase I. In phase 2, the recovered cell pool from phase 1 was treated with 1000 nM MTX and 50 μg/ml Puromycin.

After producing stable cell lines, they were cultured in RPMI 1640 with 10% fetal bovine serum (Gibco, Life Technologies, NY, USA), 100 U/mL penicillin, and 100 μg/mL streptomycin (incubated at 37˚ C, 5% $CO_2$, and 95% humidity) until the cell confluency reached 90%. Then the media was replaced with CD OptiCHO medium containing 5% efficient Feed C +AGT supplement and 3 g/L glucose and were incubated at 30˚ C in a humidified incubator with 8% $CO_2$. Twenty days later, the cell supernatants were collected and the protein concentration was determined by a homemade Sandwich ELISA assay. Briefly, the 96-well ELISA plate (PolySorp, NUNCTM, Denmark) was coated with sheep anti-human IgG antibody (at 1:3000 dilution, Sigma, Germany) and incubated overnight at 4˚C and blocked with 300 μL of PBS supplemented with 0.05% Tween 20 (Bio-Rad) and 3% nonfat skim milk (Sigma). Following that, samples and standards were prepared to be added (100 μL) to the coated plate wells and incubated at 37˚ C for 1.5 hour. Seven concentrations (100, 50, 25, 12.5, 6.25, 3.125, and 1.56 ng/mL) of a commercially available antibody was used to create a standard curve. Proteins were detected using a horseradish peroxidase (HRP)-goat anti-human IgG antibody (at a 1:10,000 dilution, Sigma). TMB substrate (Invitrogen) was added to the plate wells, and the peroxidase/TMB reaction was stopped with HCL 1N after 15 minutes. Plates were read at 450 nm on an anthos 2020 microplate reader (Biochrom, United Kingdom)."

In the next stage, Fc fusion proteins were purified using the AKTA pure FPLC system (GE Healthcare) and the HiTrap protein G HP column (GE Healthcare, Sweden).

## Physicochemical analysis

Purified Fc fusion proteins were separated by 12.5% SDS-PAGE, and stained using Coomassie Brilliant Blue (BioRad/UK). For the Western blot analysis, samples were applied to 12.5% SDS-PAGE and transferred to a polyvinylidene difluoride (PVDF) (Thermo Fisher Scientific, USA) membrane by Trans-Blot Turbo Blotting System (BioRad, CA). After blocking the PVDF membrane with 1X PBS, 5% Skim milk, and 0.15% Tween 20, blots were incubated for 1 h. at 37˚ C with goat anti-human IgG (Fc specific)-peroxidase antibody (diluted 1:60000) (Sigma, Germany). Detection of peroxidase activity was performed by Enhanced Chemiluminescence (ECL) Reagents (Bio-Rad, USA). The immunoblot was imaged by ChemiDoc™ MP System (Bio-Rad, USA) and was analyzed with Image Lab™ Software.

## *In vitro* cell line generation

Stable cell lines were generated to evaluate *in vitro* binding capacity of anti-PD-1, CD80-Fc, and 4-1BBL-Fc proteins. For this purpose, total RNA was extracted from anti-CD3/CD28

activated human PBMCs, and the first-strand cDNA was synthesized by Thermo cDNA synthesis kit (Thermo Scientific, USA). Human PD-1 (hPD-1), CTLA-4 (hCTLA-4), CD28 (hCD28), and 4-1BB (h4-1BB) coding genes were amplified by PCR using specific primers and separately inserted into pCHO1.0 expression vector. As described earlier, these constructs were transfected to CHO-K1 by electroporation, and then stable cell lines were produced by puromycin/MTX strategy. The expression of these proteins on the surface of transfected CHO-K1 cells was screened by flow cytometry using anti-PD-1-PE (BioLegend™, USA), anti-CD28-PE (BD Pharmingen™, USA), anti-CTLA4-PE (BioLegend), and anti-4-1BB-PE (BioLegend™, USA) antibodies.

## Flow cytometry analysis

The binding capacity of anti-PD-1, CD80-Fc, and 4-1BBL to recombinant stable cell lines (CHO-K1/hPD-1, CHO-K1/hCD28, CHO-K1/hCTLA4, and CHO-K1/h4-1BB) were determined by indirect flow cytometry. Briefly, $2.5 \times 10^5$ cells were treated with different concentrations (0, 0.2, 1, 5, 10 μg/mL) of purified Fc fusion proteins and incubated for 45 min. at 4˚C. The cells were washed with 2 mL of staining buffer (1X PBS buffer containing 2% fetal bovine serum). Cells were stained with FITC mouse anti-human IgG (BD, USA) for 30 min. at 4˚ C. After washing, cells were analyzed by flow cytometry using a FACS Calibur Flow cytometer (BD, USA) and FlowJo 7.6.2 software (FlowJo, LLC, CA).

## Functional analysis

Fresh whole peripheral blood was prepared from healthy human donors, and PBMCs were isolated using Ficoll density gradient separation. Donors were male and female, aged 25 to 45 years. Notably, each PBMC donor signed informed consent prior to blood collection. All procedures were verbally and in writing explained to them prior to signing the form, and all questions were fully answered. Also, the study protocols were reviewed and approved by the research ethics committee of Shiraz University of Medical Sciences (Approval ID: IR.SUMS. REC.1398.938). None of the PBMCs donors were from a vulnerable population, and all donors or next of kin provided written informed consent that was freely given.

**Mixed lymphocyte reaction (MLR).**   The MLR was employed to determine the effect of anti-PD-1, CD80-Fc, and 4-1BBL-Fc proteins and their combinations on the lymphocytes' proliferation. To do this, the γ-irradiated PBMCs (30 Gray) [16], as stimulator cells (irradiation prevents the proliferation of stimulator cells by inhibiting DNA replication but has no effect on transcription, preserving their ability to induce allogeneic T cells proliferation [16, 17]), were co-cultured with PBMCs from 6 random allogeneic donors as responder cells at a ratio of 1:1 in the presence and absence of anti-PD-1, CD80-Fc, and 4-1BBL-Fc proteins. The assay was carried out in RPMI 1640 supplemented with 10% AB human serum in a total volume of 250 μl. Before co-culturing, the responder cells were stained with CellTrace™ CFSE Cell Proliferation kit (Invitrogen) according to the manufacturer's instruction. The 3 μg/mL anti-PD-1, 5 μg/mL CD80-Fc, and 1 μg/mL 4-1BBL-Fc proteins were added to the culture. Coupled with this, PHA (Gibco®) in a suboptimal dose of 1:1000 (v/v) was used as a positive control to ensure that the system works properly. After five days, the proliferation of T lymphocytes was monitored by flow cytometry.

**Cytotoxicity assay.**   The cell-mediated cytotoxicity was assessed by 7-AAD/CFSE fluorescent probes. This study was performed in two stages: in the first stage (pre-activation), γ-irradiated PBMCs (donor A) and PBMCs (donor B) were co-cultured at a ratio of 1:1 in the presence and absence of 3 μg/mL anti-PD-1, 5 μg/mL CD80-Fc, and 1 μg/mL 4-1BBL-Fc proteins, and their combinations. After four days, cells were collected and centrifuged at 350 g for

5 min. Cells were resuspended and counted as effector cells for the second stage. Then the non-irradiated PBMCs from donor A were labeled by CFSE as target cells. After that, effector cells and CFSE-labelled target cells were co-cultured at 1:1 and 10:1 ratios. Four hours later, cells were collected and centrifuged at 350g for 5 min. To evaluate the dead target cells, 7-AAD (BioLegend) were added to the cells and incubated for 15 min. in the dark at 4° C. The cell-mediated cytotoxicity was analyzed using flow cytometry.

**Cytokine secretion analysis.** For cytokine production assay, the supernatants were collected on day 4 after an allogeneic MLR. Detection and measurement of cytokines were performed with a LEGENDplex™ Human Th Cytokine Panel (Biolegend, USA) according to the manufacturer's instructions.

## Statistical analysis

Statistical analyses were conducted using one-way analysis of variance (ANOVA) and two-way ANOVA (only for estimating the data in Fig 5A), with Dunn's method of multiple comparisons against the control condition by Prism software version 8 (GraphPad, La Jolla, CA, USA). The data were expressed as mean ± SEM. A p-value of $<0.05$ was considered as a significant difference.

# Results

## Production of anti-PD-1, CD80-Fc, and 4-1BBL-Fc proteins

Anti-PD-1 scFv, CD80 ECD, and 4-1BBL ECD were genetically fused to Fc region of IgG4 with an *S228P* mutation. After transfection of CHO-K1 with recombinant pCHO1.0 vectors, the secreted proteins were purified by protein G chromatography and concentrations of purified proteins were determined by Sandwich ELISA (23.9, 21.1, 31 μg/mL, respectively). Integrity of the purified protein was determined by western blotting under reducing conditions. The western blot results confirmed the Fc fusion proteins' identity and expected molecular weight (around 55 kDa) (Fig 1).

## Binding to native PD-1, CD28, CTLA-4, and 4-1BB receptors

The cell-based binding assay was performed on engineered CHO-K1 (CHO-K1/hPD-1, CHO-K1/hCD28, CHO-K1/hCTLA4, and CHO-K1/h4-1BB) by flow cytometry. First, the expression of the checkpoint receptors (PD-1, CD28, CTLA-4, and 4-1BB) on engineered CHO-K1 was investigated using commercial antibodies by flow cytometry. Then, the various concentrations of purified anti-PD-1, CD80-Fc, and 4-1BBL-Fc (0, 0.2, 1, 5, and 10 μg/mL) were assessed to discover the non-specific binding and the minimum detectable concentration. The results revealed that the binding of anti-PD-1, CD80-Fc, and 4-1BBL-Fc proteins were detected in all the concentrations, even 0.2 μg/mL, and the value remarkably increased with the increasing concentrations of each protein (as a result of their special capability to bind their targets) (Fig 2).

## T cell Proliferation

An allogeneic mixed lymphocyte reaction was utilized to reveal induction of proliferation in T cells. To do this, PBMCs from 6 healthy donors were labeled with 5 μM CFSE and used as responders. Other donor PBMCs were γ -irradiated at a dose of 30 Gy and used as stimuli. Responder and stimulator cells were cultured in a ratio of 1: 1 for 5 days, and the proliferation index was examined by flow cytometry.

In the first step, the sub-optimal concentration of each Fc fusion molecules was calculated by examining the concentration of 0–20 μg/mL. The results showed that T cells proliferated

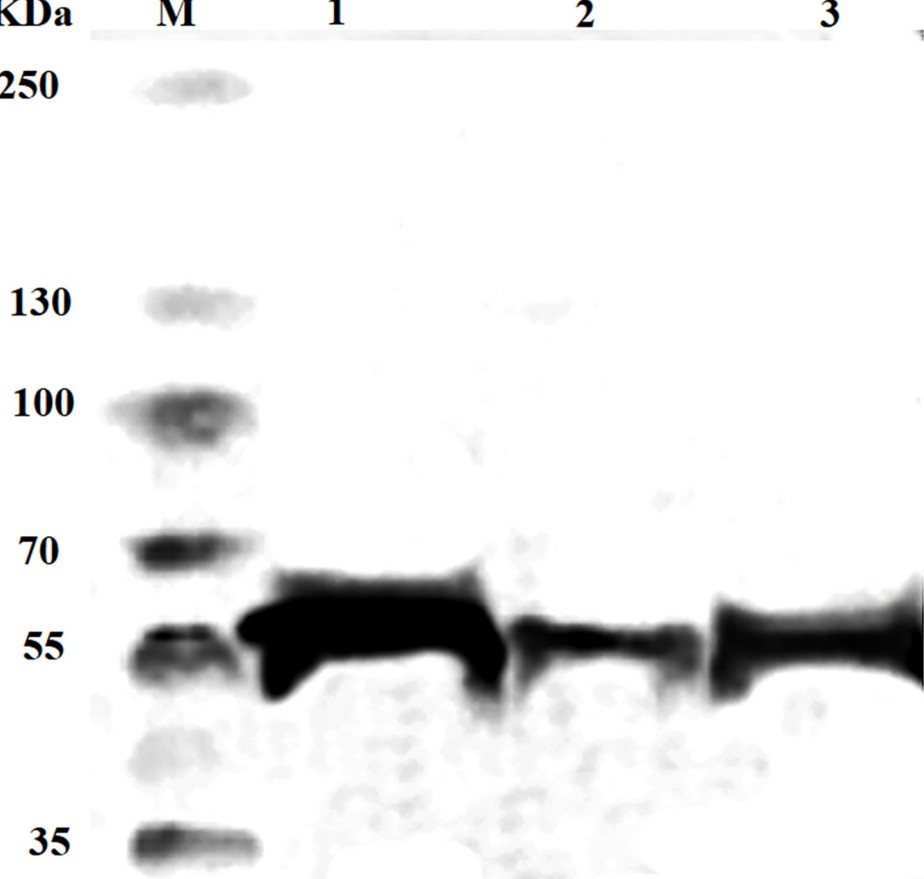

**Fig 1. Western blot analysis of purified Fc fusion proteins by probing with goat anti-human IgG (Fc specific)-peroxidase antibody under reducing conditions.** M: Protein ladder, Lane 1: anti- PD-1, Lane 2: CD80-Fc, and Lane 3: 4-1BBL-Fc.

sub-optimally at concentrations of 3 μg/mL for anti-PD-1, 5 μg/mL for CD80-Fc, and 1 μg/mLfor 4BBL-Fc, therefore these concentrations were selected for combination treatment conditions.

The M1 and M2 population in each histogram plot were considered to evaluate T cell proliferation (Fig 3A). Indeed, the M2 is a subset of the M1 population that contains cells with a high division rate. The results of the statistical significance (p-value of <0.05) were the same for both populations (Fig 3B).

The proliferation of the responder-only condition was lower than that of untreated condition (responder/stimulator), confirming the stimulatory effect of γ-irradiated PBMCs on these cells. Furthermore, the MLR results revealed that combinations of anti-PD-1/CD80-Fc (36.7%), CD80-Fc/4-1BBL-Fc (38.25%), and anti-PD-1/CD80-Fc/4-1BBL-Fc (40.4%) significantly increased T cells proliferation compared to the untreated condition (20%) (Fig 3B). Other treatments also increased T cell proliferation but these increases were not statistically significant.

## Cytokine production measurement

Four days after allogeneic MLR, the supernatants were collected and the secretion of cytokines was measured using a LEGENDplex™ bead-based multiplex kit. The results showed that IFN-γ,

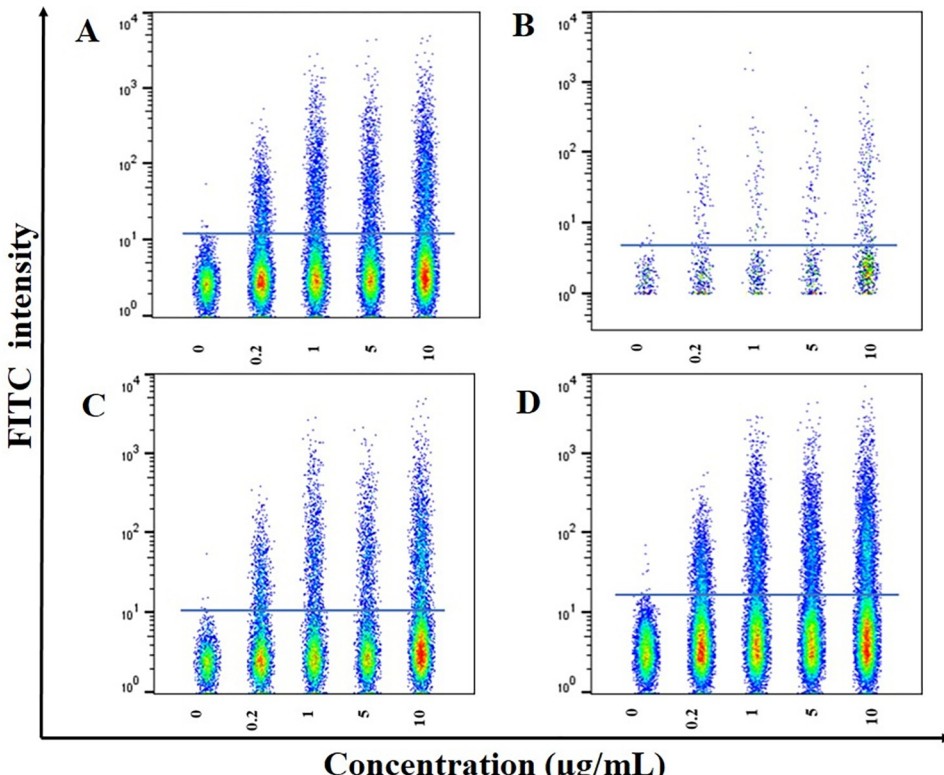

**Fig 2. The flow cytometry binding analysis of Fc fusion proteins with engineered CHO-K1 cells in 0, 0.2, 1, 5, and 10 μg/mL concentrations.** Binding capacity was detected by FITC mouse anti-human IgG and using stable cell lines engineered **A)** human PD-1 (CHO-K1/hPD-1) **B)** human CD28 (CHO-K1/hCD28) **C)** human CTLA4 (CHO-K1/hCTLA4), and **D)** human 4-1BB (CHO-K1/h4-1BB).

TNF-α, and interleukin 2 (IL-2) levels significantly increased following mono treatments and CD80-Fc/4-1BBL-Fc combination treatment compared with the untreated condition. But, anti-PD-1/ 4-1BBL-Fc and anti-PD-1/CD80-Fc/4-1BBL-Fc combinations could significantly induce the secretion of IFN-γ+IL-2 and IFN-γ+TNF-α, respectively (Fig 4A–4C). IL-10, IL-17A, IL-17F, IL-21, IL-22, IL-4, IL-5, IL-9, and IL-13 did not change significantly in any of the mono and combination treatments (Fig 4D–4F); however, some of them are not presented here.

## Cytotoxicity assay

The effector cells and CFSE labeled target cells were co-cultured at 1:1 and 10:1 ratios (Fig 5A), and the dead target cells were detected by a 7AAD probe. At 10:1 ratio, anti-PD-1 (17%), 4-1BBL-Fc (19.2%), anti-PD-1/CD80-Fc (18.6%), anti-PD-1/4-1BBL-Fc (21%), CD80-Fc/4-1BBL-Fc (18.5%), and anti-PD-1/CD80-Fc/4-1BBL-Fc (17.3%) significantly killed the target cells compared with the untreated condition (7.8%) (Fig 5B). The lowest increase was achieved after treatment with anti-PD-1 and the highest increase after co-treatment with anti-PD-1/4-1BBL-Fc.

## Discussion

It has been shown that the immunosuppressive pressure of the tumor cells on T cells may induce exhaustion, dysfunctional status and highlights the importance of developing strategies

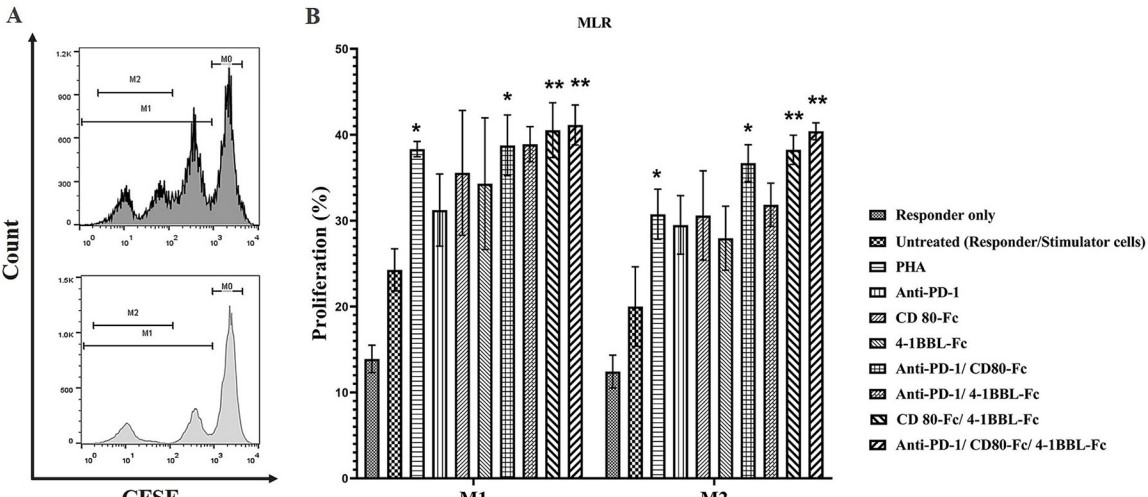

**Fig 3. T cell proliferation induction of anti-PD-1, CD80-Fc, and 4-1BBL-Fc proteins in mono and combination treatments.** For an allogeneic MLR, responder cells from 6 healthy donors were stained with CFSE and mixed with γ-irradiated PBMCs at a 1:1 ratio. PBMCs were harvested after 5 days and analyzed by flow cytometry. **(A)** Histogram plots depict examples of two populations used in this study for MLR analysis: M1 (divided) and M2 (highly divided) populations of untreated (Light gray) and PHA conditions (dark gray). M2 is a subset of the M1 population that contains cells with a high division rate. **(B)** Percentage of CFSE intensity was measured in different conditions and statistical analysis was performed using a one-way ANOVA test (*$p \leq 0.05$, **$p \leq 0.01$). The column height represents the average percentage of the CFSE- labeled cells and error bars are ± SEM.

to restore T cell functions by increasing their anti-tumor activity. The exact molecular mechanisms of T cell dysfunction are not well elucidated; however, the role of the immune checkpoint molecules have been well established as the main key regulators of the immune responses. Accordingly, a set of these modalities has been introduced to restore T cell function in tumor microenvironments, some of which are currently used in clinics. Despite the

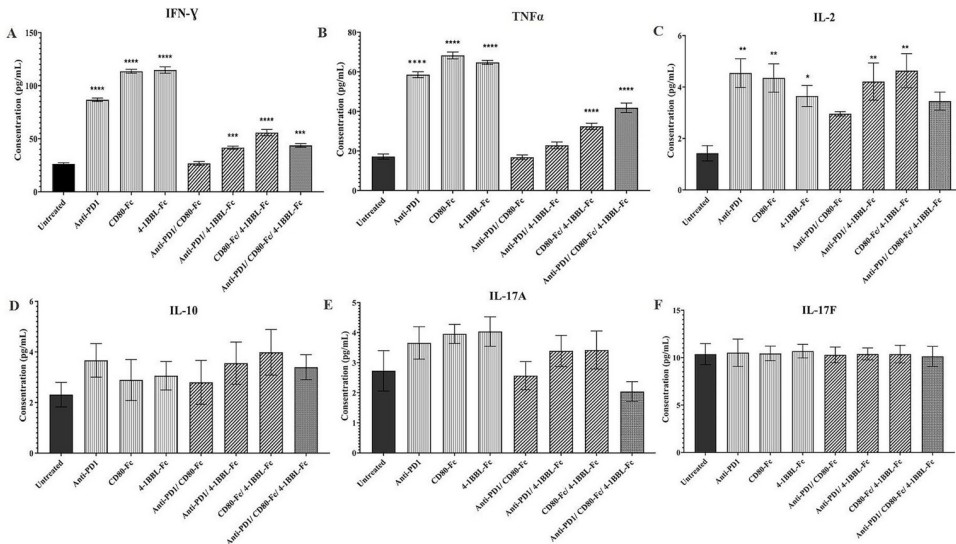

**Fig 4. Potential of mono and combination treatments of anti-PD-1, CD80-Fc, and 4-1BBL-Fc proteins on cytokine release.** An allogeneic MLR was performed and supernatants were harvested on day 4 for cytokine analysis in triplicates. This figure shows concentration of 6 cytokines: (A) IFN-γ (B) TNF-α (C) IL-2 (D) IL-10 (E) IL-17A (F) IL-17F. Data is reported as mean ± SEM (*$p \leq 0.05$, **$p \leq 0.01$, ***$P \leq 0.0001$, ****$P \leq 0.0001$).

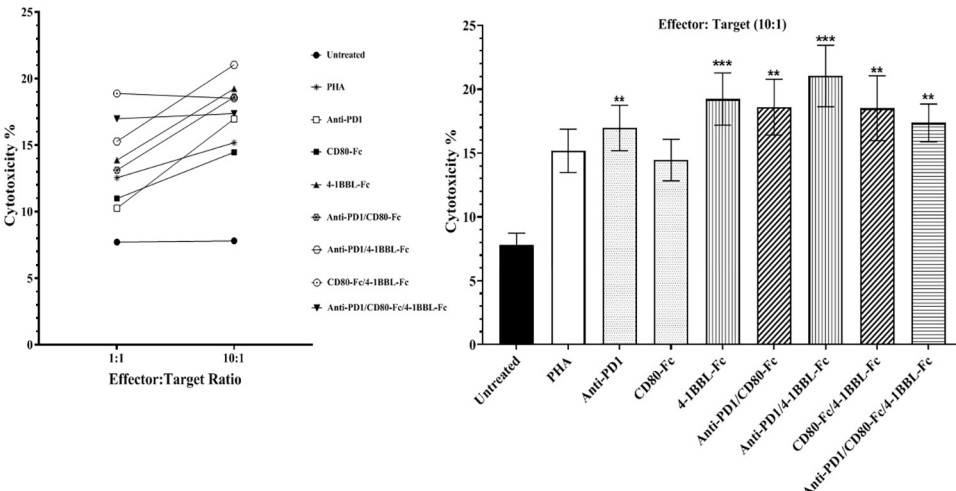

**Fig 5. T cells cytotoxicity activity in the presence and absence of anti-PD-1, CD80-Fc, and 4-1BBL-Fc proteins in two types of treatment, namely, mono and combination.** CFSE-target cells and effector cells from 6 healthy donors were co-cultured **(A)** at 1:1 and 10:1 ratios (two-way ANOVA was applied for the data collected) **(B)** At 10:1 ratio, all of the conditions were compared with the untreated using one-way ANOVA. (*p≤ 0.05, **p≤0.01, ***P≤0.0001). The column height represents the average percentage of the 7-AAD-positive CFSE-target cells and error bars are ± SEM.

preliminary promising results of the immune checkpoint therapies, only 15 to 60 percent of patients gain benefits from these drugs based on the latest information [1]. Two intrinsic or extrinsic factors have been described that are responsible for this observation [18]. Tumor may induce drastic changes in molecular pathways, the process of gene expression, DNA damage response, and cell signaling known as intrinsic factors. This can be exemplified by loss of expression of PD-L1 during the pembrolizumab administration [19]. T cell activation process also alters during anti-tumor immunity due to pressure induced by TME which is called an extrinsic factor of resistance against immune checkpoint inhibitors. The need of overcoming these resistance factors is growing by stimulating multiple stimulatory and inhibitory molecules of T cells at the same time. In this regard, we produced and evaluated the effectiveness of each of the anti-PD-1, CD80-Fc, and 4-1BBL-Fc molecules along with their combination on improving the antitumor activity of T cells.

The Fc fusion proteins were produced in CHO-K1 cells and purified by protein G. The concentrations of purified anti-PD-1, CD80-Fc, and 4-1BBL-Fc proteins were determined by Sandwich ELISA assay (23.9, 21.1, 31 μg/mL, respectively). The western blotting results confirmed our proteomic identification of the anti-PD-1, CD80-Fc, and 4-1BBL-Fc proteins (Fig 1). Furthermore, the binding capacities of the anti-PD-1, CD80-Fc, and 4-1BBL-Fc molecules to their ligands were confirmed using CHO-K1 stable cell lines engineered to express high levels of hPD-1, hCD28, hCTLA-4, and h4-1BB receptors. The interaction was detectable even in low concentrations (Fig 2).

In order to analyze the MLR results, M1 (divided) and M2 (highly divided) populations were considered. The statistically significant results (p-value of 0.05) of the two populations were the same, suggesting that the M2 population is primarily responsible for the proliferation results in this study. According to the MLR assay results, T lymphocyte proliferation increased significantly in the presence of anti-PD-1/CD80-Fc, CD80-Fc/4-1BBL-Fc, and anti-PD-1/CD80-Fc/4-1BBL-Fc combinations compared to untreated conditions (1.83-, 1.91-, and 2.02-fold respectively) (Fig 3B).

MLR results revealed that CD80-Fc, which was present in all three mentioned combination treatments (anti-PD-1/CD80-Fc, CD80-Fc/4-1BBL-Fc, and anti-PD-1/CD80-Fc/4-1BBL-Fc), played an important role in T cell proliferation. The CD80 has shown to have a variety roles in TME, such as promoting co-stimulatory signals through binding to CD28, and CTLA4, additionally, it is demonstrates that CD80 binds to PD-L1 and restores T cell activation by preventing PD-1/PD-L1 interaction [4, 5]. Concurrent stimulation of TCR and CD28 in the surface of naive T cells increase threshold of T cells priming by enhancing sensitivity of the T cell to antigen receptor engagement through fluctuating calcium response, inducing anti-apoptotic proteins expression including Bcl-XL resulted in increasing T cells proliferation at otherwise low concentrations of target antigen [20]. Our designed—MLR could mimic antigen recognition and our result confirmed the role of CD80-Fc to trigger CD28 on T cells proliferation. However, the data of applying "superagonistic" anti-CD28 in TCR independent manner, suggest that T cells in TME regardless of the TCR engagement maybe benefits from CD80-Fc [21].

An allogeneic MLR in a previous study confirmed the role of anti-PD-1 on T cell proliferation. According to this, PD-1 expression on regulatory T cells (Tregs) allowed nivolumab to suppress $CD4^+CD25^+$ Treg proliferation and, as a result, restore $CD4^+$ T cells and production of IFN-γ [13]. Indeed, PD-1 blockade can increase effector T cells' proliferation by augmenting the resistance of $CD8^+$ to Tregs suppression and help to overcome the inhibitory effect of Tregs [22]. Selby et al. demonstrated that nivolumab, on its own and in combination with ipilimumab, increased T cell proliferation and IFN-γ production in concentrations greater than 0.5 μg/mL [23].

Moreover, 4-1BB is overexpressed on the surface of activated T and NK cells after the administration of PD-1/PD-L1 blocking agents. On the other hand, the 4-1BB/4-1BBL interaction induces the primary and secondary responses of $CD4^+$ and $CD8^+$ T cells independently of CD28 signals. The co-stimulatory signals of 4-1BB are triggered by activating NF-κB, c-Jun, and p38 MAPK pathways, which leads to transcription of several genes involved in production of IL-2 and IFN-γ [24, 25]. Also, 4-1BB prevents activation-induced cell death (AICD) and increases anti-apoptotic signals in T cells, and subsequently leads to induce differentiation (to effector and memory cells) and proliferation of T cells [26]. Therefore, the employment of 4-1BB agonists may augment the antitumor activity of these immune cell populations. Previous studies have shown the stimulatory effect of 4-1BBL on T cell proliferation and immune response capacity. In addition, according to the MLR assay results, a combination of anti-4-1BB/anti-PD-1 antibodies and anti-PD-1/anti-4-1BB bispecific antibody significantly enhanced T cell proliferation in comparison with monotherapy with anti-4-1BBL and anti-PD-1 antibodies [27].

In this study, cytokine production of the PBMCs was evaluated in the presence of individual proteins and their combination. According to our findings, combination treatments generated the most effective and powerful responses, although the production level of cytokines that affect the cellular response (including IFN-γ and TNF-α) was higher in mono treatments (Fig 4A–4C). It seems that an optimal dose of cytokines is needed to have the best efficacy of T cells activity induction, which, in the authors' study-case, may well have been made available by combination therapy. Additionally, due to the short half-life and degradation of cytokines, the best time points for evaluating peak concentrations in this study for IFN-γ and TNF-α may have been missed. In addition, the increased proliferation and cytotoxicity associated with combination treatment may have occurred as a result of the induction of other cellular mechanisms. Interestingly, none of the mono and combination treatments induced the production of IL-10, IL-17A, IL-17F, and IL-22 cytokines (Fig 4D–4F).

Our cytotoxicity assay results revealed that anti-PD-1 (17%), 4-1BBL-Fc (19.2%), anti-PD-1/CD80-Fc (18.6%), anti-PD-1/4-1BBL-Fc (21%), CD80-Fc/4-1BBL-Fc (18.5%), and anti-PD-

1/CD80-Fc/4-1BBL-Fc (17.3%) significantly killed the target cells compared with the untreated condition (7.8%). Indeed, all the combination treatments were effective in the target cell killing. Higher cytotoxicity was observed in the presence of anti-PD-1/4-1BBL-Fc (2.6-folds greater than in the untreated conditions), which could be attributed to the expansion of T cell effector subsets, particularly CD8$^+$ T cells (Fig 5B). In a 2015 study, it was reported that a combination of anti-4-1BB/anti-PD-1 enhanced the CD8$^+$/Treg ratio [28] which is in favor of anti-tumor immune response. By applying anti-PD-1/4-1BBL-Fc as a combination, they may exert a synergistic effect on T cells function by reducing the inhibitory signals and increasing the activatory ones through the blockade of PD-L1 and the stimulation of 4-1BB receptors, respectively. The capacity of this mechanism has sparked a lot of interest in applying it in combination therapy. Indeed, the anti-4-1BB increases the functional activity of tumor-specific CD8$^+$ CTLs by inducing the expression of T-box transcription factor Eomesodermin (Eomes) [28–30]. Furthermore, treatment of murine lymphoma model with anti-4-1BB protein showed tumor rejection due to an increased cytotoxicity response of CD8$^+$ T cells by the augmentation of perforin-granzyme and FasL mechanisms [31].

Data from this research collectively support stronger cytotoxic effects of combination therapy than monotherapy for cancer treatment. A finding in line of the effectiveness of nivolumab with nivolumab plus ipilimumab in patients with renal cell carcinoma (RCC) which has shown the superiority of combination therapy [32]. To confirm the advantage of combination therapy it has been shown that ipilimumab provides suitable conditions for nivolumab response by expressing transcription factors and predisposing the tumor environment [33]. In addition, up-regulation of the Treg chemoattractant CCL28 increases Treg population and disrupts the desirable CD4:CD8 ratio, which ipilimumab is able to reduce Tregs by ADCC effect [34]. Likewise, in this study, the various effect of anti-PD-1, CD80-Fc, and 4-1BBL-Fc molecules on the cellular signaling pathways can lead to providing the conditions for an effective response from each molecule and enhancing the immune response in combination therapy. 4-1BBL-Fc protein induces PD-1 expression on effector T cells, which in turn can lead to resistance to 4-1BBL-Fc therapy in tumors [28]. Anti-PD-1/anti 4-1BB combination treatment increases the expression of PD-L1 on tumor cells by producing IFN-γ, thus providing a rational for simultaneous treatment of tumor with these two proteins [4, 28].

Studies have shown that CD28 superagonist antibodies bind to the C"D loop of the CD28 Ig-like domain and activate naïve T cells in the absence of TCR stimulation. These superagonist antibodies lead to the release of many cytokines (cytokine storm) and induce Treg cells expansion in the tumor environment [35]. In contrast, CD80-Fc binds to the natural binding region of the CD28 molecule ("MYPPPY" conserved motif within the IgV-like domain [36]) and is not capable of spontaneously cytokine release. CD80-Fc also blocks PD-1/PD-L1 interaction by binding to PD-L1, which is expressed by activated T cells in the TME. Horn et al. have shown that binding of soluble CD80 to CTLA-4 does not limit T cell activation and helps to trigger the CD28 stimulatory signal by inhibiting competition for endogenous CD80 [6]. According to these studies, CD80-Fc may even be effective in treating anti-PD-1 or anti-CTLA-4 -resistant patients and in our study its combination with 4-1BBL-Fc (Figs 3 and 5) enhanced immune response against tumor cells.

## Conclusion

Current findings suggest that the various combinations of treatments with produced anti-PD-1, CD80-Fc, and 4-1BBL-Fc proteins reduce inhibitory signals while increasing activatory signals, potentially lowering the risk of tumor resistance and improving treatment efficacy. Because of the expression of different molecular markers (including PD-L1) and different

tumor characteristics, each of the combination treatments presented in this study may be a promising proposal for treatment of different types of cancers. Another viable explanation for this efficacy improvement following co-treatment (aside from breaking down patient resistance to monotherapy) could be the presence of multiple T cell subsets with varying levels of co-inhibitory and co-stimulatory receptors on their cell surface [5, 37, 38]. Despite the advantages of this approach, caution should be exercised in designing and selecting the combination of proteins and their optimum doses to accomplish the most effective treatment.

## Supporting information

**S1 Raw images. Original image of western blot.**
(PDF)

**S1 File. Minimal dataset.**
(PDF)

## Author Contributions

**Conceptualization:** Elina Kaviani, Ahmad Hosseini, Abbas Ghaderi, Amin Ramezani.

**Data curation:** Elina Kaviani, Amin Ramezani.

**Formal analysis:** Elina Kaviani, Ahmad Hosseini, Abbas Ghaderi, Amin Ramezani.

**Investigation:** Elina Kaviani, Ahmad Hosseini, Elham Mahmoudi Maymand, Abbas Ghaderi, Amin Ramezani.

**Methodology:** Elina Kaviani, Ahmad Hosseini, Mani Ramzi, Abbas Ghaderi, Amin Ramezani.

**Project administration:** Abbas Ghaderi, Amin Ramezani.

**Software:** Elina Kaviani, Mani Ramzi, Amin Ramezani.

**Supervision:** Abbas Ghaderi, Amin Ramezani.

**Validation:** Elina Kaviani, Ahmad Hosseini, Abbas Ghaderi, Amin Ramezani.

**Visualization:** Elina Kaviani, Ahmad Hosseini, Mani Ramzi.

**Writing – original draft:** Elina Kaviani.

**Writing – review & editing:** Elina Kaviani, Ahmad Hosseini, Elham Mahmoudi Maymand, Mani Ramzi, Abbas Ghaderi, Amin Ramezani.

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
