## [Decision Letter · Decision Letter 0]

11 Jul 2022

PONE-D-22-10263Triggering of lymphocytes by CD28, 4-1BB and PD-1 checkpoints to enhance the immune response capacitiesPLOS ONE

Dear Dr. Ramezani,

Thank you for submitting your manuscript to PLOS ONE. After careful consideration, we feel that it has merit but does not fully meet PLOS ONE’s publication criteria as it currently stands. Therefore, we invite you to submit a revised version of the manuscript that addresses the points raised during the review process.Both reviewers have taken time out to review your manuscript, please read each comment carefully and revise your manuscript according.

authors should followProvide specific feedback from your evaluation of the manuscriptPlease ensure that your decision is justified on PLOS ONE’s publication criteria and not, for example, on novelty or perceived impact.

We look forward to receiving your revised manuscript.

Kind regards,

Suzie Chen

Academic Editor

PLOS ONE

Journal Requirements:

3. Please provide the following information regarding PBMC donors for the analysis in your study.

1. Please provide the source(s) of the tissue/organs used in the study, including the institution name and a non-identifying description of the donor(s).

2. Please state in your response letter and ethics statement whether this study involved any vulnerable populations; for example, tissue/organs from prisoners, subjects with reduced mental capacity due to illness or age, or minors.

- If a vulnerable population was used, please describe the population, justify the decision to use tissue/organ donations from this group, and clearly describe what measures were taken in the informed consent procedure to assure protection of the vulnerable group and avoid coercion. 

- If a vulnerable population was not used, please state in your ethics statement, “None of the transplant donors was from a vulnerable population and all donors or next of kin provided written informed consent that was freely given.”

3. In the Methods, please provide detailed information about the procedure by which informed consent was obtained from organ/tissue donors or their next of kin. In addition, please provide a blank example of the form used to obtain consent from donors, and an English translation if the original is in a different language.

4. Please indicate whether the donors were previously registered as organ donors. If tissues/organs were obtained from deceased donors or cadavers, please provide details as to the donors’ cause(s) of death.

5. Please provide the participant recruitment dates and the period during which the sample collection were done (as month and year).

6. Please discuss whether medical costs were covered or other cash payments were provided to the family of the donor. If so, please specify the value of this support (in local currency and equivalent to U.S. dollars)

Reviewers' comments:

Reviewer's Responses to Questions

**Comments to the Author**

1. Is the manuscript technically sound, and do the data support the conclusions?

Reviewer #1: Partly

Reviewer #2: Partly

2. Has the statistical analysis been performed appropriately and rigorously? 

Reviewer #1: Yes

Reviewer #2: Yes

3. Have the authors made all data underlying the findings in their manuscript fully available?

Reviewer #1: Yes

Reviewer #2: No

4. Is the manuscript presented in an intelligible fashion and written in standard English?

Reviewer #1: Yes

Reviewer #2: Yes

5. Review Comments to the Author

Reviewer #1: The manuscript entitled: “Triggering of lymphocytes by CD28, 4-1BB and PD-1 checkpoints to enhance the immune response capacities” investigated co-stimulatory effects of three an anti-PD-1, a CD80 extracellular domain (ECD)-Fc, and a 4-1BBL extracellular domain (ECD)-Fc fusion proteins in vitro, on T cells’ proliferation, cytotoxicity, and cytokines secretion.

Overall, the survey is of interest to scientists in the field, however the manuscript is not well-organized and some data are difficult to follow.

There are several major concerns with the study presented by the authors-

methods:

the authors should provide The accession number of anti-PD-1, , CD80 ECD , and 4-1BBL ECD.

E 92-3: “production of anti-92 PD-1 scFv–Fc, CD80-Fc, and 4-1BBL-Fc proteins in the cell culture supernatant were evaluated by sandwich ELISA (Enzyme-Linked Immunosorbent Assay)…” the features of the kit should mention.

Line 95: increasing levels of puromycin (Gibco) and MTX…” what's the meaning of increasing MTS

Line 199: “To do this, PBMCs from 6 healthy donors were labeled with 5 μM CFSE and used as

responders.” The statement should be clarified and why γ-irradiated PBMCs used as stimuli and how it was confirmed.

Why only 10:1 ratios for Cytotoxicity assay presented.

Figure 1 are not clear and should be clarified.

Line 196 The results of T cell Proliferation should be explained in detail in discussion also the data are difficult to follow.

Reviewer #2: 1、For the discussion part, please discuss the following five parts: "production of anti-PD-1, CD80 FC and 4-1bbl-fc protein", "binding to natural PD-1, CD28, CTLA-4 and 4-1BB receptors", "T cell proliferation", "cytokine production assay" and "cytotoxicity test".

2、Add the content of conclusion.

3、Supplement the data of figure 3B, figure 4 and figure 5 in the supplementary materials.

4、Abstract- rewrite. It is hard to follow the ideas, the results and conclusion.

6. PLOS authors have the option to publish the peer review history of their article (what does this mean?). If published, this will include your full peer review and any attached files.

Reviewer #1: **Yes: **Amir Ghaemi

Reviewer #2: No

---

## [Author Response · Author response to Decision Letter 0]

9 Aug 2022

Our thanks to the reviewers and the editor for useful comments and suggestions on our manuscript. We have adopted the suggestions and modified the manuscript accordingly. We think that the manuscript has been greatly improved by these revisions and we hope that you will now find it suitable for publication in PLOS ONE. Our point-by-point detailed responses to comments are listed on the following pages.

Response to Editor:

Editor: Please ensure that your manuscript meets PLOS ONE's style requirements, including those for file naming.

Authors: The manuscript was prepared in the correct format of PLOS ONE style

Editor: PLOS ONE now requires that authors provide the original uncropped and unadjusted images underlying all blot or gel results reported in a submission’s figures or Supporting Information files.

Authors: The original uncropped and unadjusted image of Western blot (Fig 1) was provided and reported in a Supporting Information files.

Editor: Please provide the following information regarding PBMC donors for the analysis in your study:

• Please provide the source(s) of the tissue/organs used in the study, including the institution name and a non-identifying description of the donor(s).

Authors: Using human PBMCs was approved by the research ethics committee of the Shiraz University of Medical Sciences (Approval ID: IR.SUMS.REC.1398.938). PBMCs were isolated from freshly peripheral blood of healthy donors. Details of donors was include sex: male and female, age range: 25-45 years with a non-vulnerable population, sampling: PBMCs were isolated from fresh blood by ficoll-gradient.

• Please state in your response letter and ethics statement whether this study involved any vulnerable populations; for example, tissue/organs from prisoners, subjects with reduced mental capacity due to illness or age, or minors.

Authors: No vulnerable population was used in this study and this sentence was added: “None of the PBMCs donors were from a vulnerable population, and all donors or next of kin provided written informed consent that was freely given.” to our ethics statement.

• In the Methods, please provide detailed information about the procedure by which informed consent was obtained from organ/tissue donors or their next of kin. In addition, please provide a blank example of the form used to obtain consent from donors, and an English translation if the original is in a different language.

Authors: A detail of informed consent from donors was added to the method section. A blank example of the form was prepared and provided.

 “Fresh whole peripheral blood was prepared from healthy donors, and PBMCs were isolated using Ficoll density gradient separation. Donors were male and female, aged 25 to 45 years, and from a non-vulnerable population. Notably, each PBMC donor signed informed consent prior to blood collection. All procedures were verbally and in writing explained to them prior to signing the form, and all questions were fully answered. Also, the study protocols were reviewed and approved by the research ethics committee of Shiraz University of Medical Sciences (Approval ID: IR.SUMS.REC.1398.938).”

• Please indicate whether the donors were previously registered as organ donors. If tissues/organs were obtained from deceased donors or cadavers, please provide details as to the donors’ cause(s) of death.

Authors: Their medical history was requested on the informed consent form, and the donors had not previously registered as organ donors. Donors were required to be healthy and free of the disorder in order to participate in our study.

• Please c and the period during which the sample collection were done (as month and year).

Authors: Donors were asked to attend a maximum of three appointments over the course of 30 days (specified on the informed consent form).

• Please discuss whether medical costs were covered or other cash payments were provided to the family of the donor. If so, please specify the value of this support (in local currency and equivalent to U.S. dollars)

Authors: The research group covered all medical costs, but the donors' families received no monetary or medical compensation for their donation of these cells.

Editor: In your Data Availability statement, you have not specified where the minimal data set underlying the results described in your manuscript can be found. PLOS defines a study's minimal data set as the underlying data used to reach the conclusions drawn in the manuscript and any additional data required to replicate the reported study findings in their entirety. All PLOS journals require that the minimal data set be made fully available.

Authors: A study's minimal data set was prepared and uploaded as Supporting Information files.

Response to Reviewers:

Reviewer #1: The authors should provide the accession number of anti-PD-1, CD80 ECD, and 4-1BBL ECD.

Authors: The accession number of anti-PD-1, CD80 ECD, and 4-1BBL ECD were added to manuscript: 

“The anti-PD-1 scFv of nivolumab (variable heavy and light chains of nivolumab were used to design anti-PD1 scFv [DrugBank, accession number: DB09035], CD80 ECD [accession number: NP_005182.1], and 4-1BBL ECD [accession number: NP_003802.1] were genetically fused to fragment crystallizable (Fc) region (Hinge-CH2-CH3) of human immunoglobulin gamma 4 (IgG4) with an S228P mutation” 

Reviewer #1: E 92-3: “production of anti-92 PD-1 scFv–Fc, CD80-Fc, and 4-1BBL-Fc proteins in the cell culture supernatant were evaluated by sandwich ELISA (Enzyme-Linked Immunosorbent Assay)…” the features of the kit should mention.

Authors: A homemade Sandwich ELISA was used in this experiment. The ELISA procedure was briefly added to this section at the request of the respected Reviewer. 

 “Twenty days later, the cell supernatants were collected and the protein concentration was determined by a homemade Sandwich ELISA assay. Briefly, the 96-well ELISA plate (PolySorp, NUNCTM, Denmark) was coated with sheep anti-human IgG antibody (at 1:3000 dilution, Sigma, Germany) and incubated overnight at 4o C and blocked with 300 µL of PBS supplemented with 0.05% Tween 20 (Bio-Rad) and 3% nonfat skim milk (Sigma). Following that, samples and standards were prepared to be added (100 µL) to the coated plate wells and incubated at 37° C for 1.5 hour. Seven concentrations (100, 50, 25, 12.5, 6.25, 3.125, and 1.56 ng/mL) of a commercially available antibody was used to create a standard curve. Proteins were detected using a horseradish peroxidase (HRP)-goat anti-human IgG antibody (at a 1:10,000 dilution, Sigma). TMB substrate (Invitrogen) was added to the plate wells, and the peroxidase/TMB reaction was stopped with HCL 1N after 15 minutes. Plates were read at 450 nm on an anthos 2020 microplate reader (Biochrom, United Kingdom).”

Reviewer #1: Line 95: increasing levels of puromycin (Gibco) and MTX…” what's the meaning of increasing MTS.

Authors: This section now contains more detailed information about the “two-phase selection strategy with MTX/ Puromycin”.

“In brief, 20 µg/mL Puromycin and 200 nM MTX were used in selection phase I. In phase 2, the recovered cell pool from phase 1 was treated with 1000 nM MTX and 50 µg/ml Puromycin”

Reviewer #1: Line 199: “To do this, PBMCs from 6 healthy donors were labeled with 5 μM CFSE and used as responders.” The statement should be clarified and why γ-irradiated PBMCs used as stimuli and how it was confirmed.

Authors: We appreciate the respected Reviewer’s insightful comment. The one-way MLR is initiated by reducing the proliferative response to a single donor (responder) by an irradiated donor (stimulator). The γ-irradiated PBMCs were cultured, and no proliferation was observed for 6 days. To confirm the stimulatory effect of PBMCs exposed to γ-irradiation, a condition was considered in which CFSE-responder cells were cultured alone in this experiment. The proliferation of this condition was lower than untreated (Responder/Stimulator). According to the Reviewer’s comment, this condition was added to the other conditions and presented in Figure 3B.

Reviewer #1: Why only 10:1 ratios for Cytotoxicity assay presented?

Authors: Great point. For the cytotoxicity assay, the effector cells and CFSE labeled target cells were co-cultured, and the results were compared at 1:1, 5:1, and 10:1 ratios. Since data of the multiple effector: target ratios are usually presented by superimposed symbols with a connecting line, first, our cytotoxicity results were plotted with connecting lines to assess the specificity of the response. As shown in the figure below, multiple treatments conditions, different ratios, and the same pattern of results make it challenging for the reader to understand at a glance. To solve this problem, we used the data with the lowest (1:1) and the highest (10:1) ratio to determine the specificity of cytotoxicity function (Fig 5A) and we spared the presentation of 5:1 ratio data. To better show the differences in the results of various treatments and the same pattern of them at all ratios, the results were displayed separately at a 10:1 ratio (Fig 5B).

Reviewer #1: Figure 1 are not clear and should be clarified

Authors: We attempted to clarify the results of Figure 1 by changing some sentences.

“Anti-PD-1 scFv, CD80 ECD, and 4-1BBL ECD were genetically fused to Fc region of IgG4 with an S228P mutation. After transfection of CHO-K1 with recombinant pCHO1.0 vectors, the secreted proteins were purified by protein G chromatography and concentrations of purified proteins were determined by Sandwich ELISA (23.9, 21.1, 31 µg/mL, respectively). Integrity of the purified protein was determined by western blotting under reducing conditions. The western blot results confirmed the Fc fusion proteins' identity and expected molecular weight (around 55 kDa) (Fig 1).”

Reviewer #1: Line 196: The results of T cell Proliferation should be explained in detail in discussion also the data are difficult to follow.

Authors: Thank you for point this out. The discussion now includes details on proliferation data: 

“According to the MLR assay results, T lymphocyte proliferation increased significantly in the presence of anti-PD-1/CD80-Fc, CD80-Fc/4-1BBL-Fc, and anti-PD-1/CD80-Fc/4-1BBL-Fc combinations compared to untreated conditions (1.83-, 1.91-, and 2.02-fold respectively) (Fig 3B). MLR results revealed that CD80-Fc, which was present in all three mentioned combination treatments (anti-PD-1/CD80-Fc, CD80-Fc/4-1BBL-Fc, and anti-PD-1/CD80-Fc/4-1BBL-Fc), played an important role in T cell proliferation.”

Reviewer #2: For the discussion part, please discuss the following five parts: "production of anti-PD-1, CD80 FC and 4-1bbl-fc protein", "binding to natural PD-1, CD28, CTLA-4 and 4-1BB receptors", "T cell proliferation", "cytokine production assay" and "cytotoxicity test".

Authors: We appreciate the reviewer’s insightful comment. The discussion part was divided to 6 parts, “the introduction and thoughts on the importance of the issue”, "binding to natural PD-1, CD28, CTLA-4 and 4-1BB receptors", "T cell proliferation and effect of each Fc fusion proteins on it", "cytotoxicity test and role of Fc fusion proteins on it", "cytokine production assay", “general conclusions from the obtained results and evidence from previous studies about it”. 

According to the comment of the respected Reviewer, the arrangement of the discussion section was changed to “the introduction and thoughts on the importance of the issue”, "production of anti-PD-1, CD80-Fc, and 4-1BBL-Fc protein", binding to natural PD-1, CD28, CTLA-4 and 4-1BB receptors", "T cell proliferation", "cytokine production assay", "cytotoxicity test", and “the main finding of our study and review of evidence from previous studies for it”.

Reviewer #2: Add the content of conclusion.

Authors: Thank you for point this out. A conclusion section was added to the manuscript.

“Current findings suggest that the various combinations of treatments with produced anti-PD-1, CD80-Fc, and 4-1BBL-Fc proteins reduce inhibitory signals while increasing activatory signals, potentially lowering the risk of tumor resistance and improving treatment efficacy. Because of the expression of different molecular markers (including PD-L1) and different tumor characteristics, each of the combination treatments presented in this study may be a promising proposal for treatment of different types of cancers. Another viable explanation for this efficacy improvement following co-treatment (aside from breaking down patient resistance to monotherapy) could be the presence of multiple T cell subsets with varying levels of co-inhibitory and co-stimulatory receptors on their cell surface [5, 39, 40]. Despite the advantages of this approach, caution should be exercised in designing and selecting the combination of proteins and their optimum doses to accomplish the most effective treatment.”

Reviewer #2: Supplement the data of figure 3B, figure 4 and figure 5 in the supplementary materials.

Authors: A file called “Minimal Data Set” was created and uploaded as Supporting Information files, containing supplementary data from figures 3B, 4 and 5. 

Reviewer #2: Abstract- rewrite. It is hard to follow the ideas, the results and conclusion.

Authors: According to the comment of the respected Reviewer, the abstract was rewritten: 

“Tumor infiltrating lymphocytes (TILs) usually become exhausted and dysfunctional owing to chronic contact with tumor cells and overexpression of multiple inhibitor receptors. Activation of TILs by targeting the inhibitory and stimulatory checkpoints has emerged as one of the most promising immunotherapy prospectively. We investigated whether triggering of CD28, 4-1BB, and PD-1 checkpoints simultaneously or alone could enhance the immune response capacity of lymphocytes. In this regard, anti-PD-1, CD80-Fc, and 4-1BBL-Fc proteins were designed and produced in CHO-K1 cells as an expression host. Following confirmation of the Fc fusion proteins' ability to bind to native targets expressed on engineered CHO-K1 cells (CHO-K1/hPD-1, CHO-K1/hCD28, CHO-K1/hCTLA4, and CHO-K1/h4-1BB), the effects of each protein, on its own and in various combinations, were assessed in vitro on T cell proliferation, cytotoxicity, and cytokines secretion using the Mixed lymphocyte reaction (MLR) assay, 7-AAD/CFSE cell-mediated cytotoxicity assay, and a LEGENDplex™ Human Th Cytokine Panel, respectively. MLR results demonstrated that T cell proliferation in the presence of the combinations of anti-PD-1/CD80-Fc, CD80-Fc/4-1BBL-Fc, and anti-PD-1/CD80-Fc/4-1BBL-Fc proteins was significantly higher than in the untreated condition (1.83-, 1.91-, and 2.02-fold respectively). Furthermore, anti-PD-1 (17%), 4-1BBL-Fc (19.2%), anti-PD-1/CD80-Fc (18.6%), anti-PD-1/4-1BBL-Fc (21%), CD80-Fc/4-1BBL-Fc (18.5%), and anti-PD-1/CD80-Fc/4-1BBL-Fc (17.3%) significantly enhanced cytotoxicity activity compared to untreated condition (7.8%). However, concerning the cytokine production, CD80-Fc and 4-1BBL-Fc alone or in combination significantly increased the secretion of IFN‐γ, TNF-α, and IL-2 compared with the untreated conditions. In conclusion, this research establishes that the various combinations of produced anti-PD-1, CD80-Fc, and 4-1BBL-Fc proteins can noticeably induce the immune response in vitro. Each of these combinations may be effective in killing or destroying cancer cells depending on the type and stage of cancer.”

---

## [Decision Letter · Decision Letter 1]

6 Sep 2022

PONE-D-22-10263R1Triggering of lymphocytes by CD28, 4-1BB and PD-1 checkpoints to enhance the immune response capacitiesPLOS ONE

Dear Dr. Ramezani,

Thank you for submitting your manuscript to PLOS ONE. After careful consideration, we feel that it has merit but does not fully meet PLOS ONE’s publication criteria as it currently stands. Therefore, we invite you to submit a revised version of the manuscript that addresses the points raised during the review process. Please submit your revised manuscript by Oct 21 2022 11:59PM. If you will need more time than this to complete your revisions, please reply to this message or contact the journal office at plosone@plos.org. Please include the following items when submitting your revised manuscript:A rebuttal letter that responds to each point raised by the academic editor and reviewer(s). You should upload this letter as a separate file labeled 'Response to Reviewers'.A marked-up copy of your manuscript that highlights changes made to the original version. You should upload this as a separate file labeled 'Revised Manuscript with Track Changes'.An unmarked version of your revised paper without tracked changes. You should upload this as a separate file labeled 'Manuscript'.

We look forward to receiving your revised manuscript.

Kind regards,

Suzie Chen

Academic Editor

PLOS ONE

Additional Editor Comments:

One of the reviewers still felt you have not addressed the concerns raised from the previous review, further revision is needed.

Reviewers' comments:

Reviewer's Responses to Questions

**Comments to the Author**

1. If the authors have adequately addressed your comments raised in a previous round of review and you feel that this manuscript is now acceptable for publication, you may indicate that here to bypass the “Comments to the Author” section, enter your conflict of interest statement in the “Confidential to Editor” section, and submit your "Accept" recommendation.

Reviewer #1: (No Response)

2. Is the manuscript technically sound, and do the data support the conclusions?

Reviewer #1: Yes

3. Has the statistical analysis been performed appropriately and rigorously? 

Reviewer #1: Yes

4. Have the authors made all data underlying the findings in their manuscript fully available?

Reviewer #1: Yes

5. Is the manuscript presented in an intelligible fashion and written in standard English?

Reviewer #1: Yes

6. Review Comments to the Author

Reviewer #1: The authors have responded to some comments made in the previous round of review and have somewhat improved the manuscript, some question about rational of different parts and organization remain.

Please authors describe how the 30 Gy was determine for γ-irradiation of PBMCs?

Please clarify the statement: (non-proliferative but potentially antigen-presenting cells)

Image (1and 4) quality need to improve

Histogram plots in Figure 3A quality is poor, Please replace the images with higher resolution images. Also the caption need clarification” (A) Histogram plots show M1 (divided) and M2 (highly 270 divided) populations of untreated (Light gray) and PHA conditions (dark gray)”

The results of T cell Proliferation ( The M1 and M2 population )should be explained in detail in discussion

7. PLOS authors have the option to publish the peer review history of their article (what does this mean?). If published, this will include your full peer review and any attached files.

Reviewer #1: **Yes: **Amir Ghaemi

---

## [Author Response · Author response to Decision Letter 1]

12 Sep 2022

We appreciate the reviewers' and editor's insightful comments, which helped strengthen our manuscript. All comments were accepted and the appropriate changes to the manuscript were made using “track changes.” We believe that these revisions have greatly improved the manuscript, and we hope that you will now consider it suitable for publication in PLOS ONE. Our point-by-point detailed responses to comments are listed on the following pages.

Response to Editors:

Editors: Please note that funding information should not appear in any section or other areas of your manuscript. We will only publish funding information present in the Funding Statement section of the online submission form. Please remove any funding-related text from the manuscript.

Authors: In response to the respected editor's comment, the “Funding information” was removed from the manuscript.

Editors: Your ethics statement should only appear in the Methods section of your manuscript. If your ethics statement is written in any section besides the Methods, please move it to the Methods section and delete it from any other section. Please ensure that your ethics statement is included in your manuscript, as the ethics statement entered into the online submission form will not be published alongside your manuscript.

Authors: The separate part titled "Ethics statement" at the end of the Conclusion section was removed and entered in the Methods section. Lines 157-160 (in the “Functional analysis” part) contained all the necessary details and explanations related to the ethics statement.

“Fresh whole peripheral blood was prepared from healthy human donors, and PBMCs were isolated using Ficoll density gradient separation. Donors were male and female, aged 25 to 45 years. Notably, each PBMC donor signed informed consent prior to blood collection. All procedures were verbally and in writing explained to them prior to signing the form, and all questions were fully answered. Also, the study protocols were reviewed and approved by the research ethics committee of Shiraz University of Medical Sciences (Approval ID: IR.SUMS.REC.1398.938). None of the PBMCs donors were from a vulnerable population, and all donors or next of kin provided written informed consent that was freely given.”

Response to Reviewer:

Reviewer 1: Please authors describe how the 30 Gy was determine for γ-irradiation of PBMCs?

Authors: Previous studies in MLR assays have shown that stimulator cells are exposed to high doses of gamma irradiation (typically 30 Gy= 3000 rad) to prevent proliferation while maintaining their ability to stimulate the proliferation and cytokine profiles of allogeneic T cells. Tourkova et al., 2001 reported that irradiated (30 Gy) and non-irradiated DCs had similar stimulatory effects. As a result, these findings suggest that irradiating stimulator cells with a 30 Gy dose had no effect on their ability to induce T cell proliferation in MLR. Therefore, in this study, 30 Gy radiation was chosen. To confirm the irradiation process efficacy, the stimulator PBMCs were stained with CFSE and cultured in the presence of PHA (1:1000 (v/v)). After 5 days, the cells were harvested and stained with 7-AAD, and a cell proliferation test was performed to confirm the accuracy of the amount of radiation used; the cells were found to be alive but unable to divide.

The relevant references have been included in the manuscript. 

• Tourkova IL, Yurkovetsky ZR, Shurin MR, Shurin GV. Mechanisms of dendritic cell-induced T cell proliferation in the primary MLR assay. Immunology letters. 2001;78(2):75-82.

• Pourfathollah A-A, Shaiegan M, Namiri M, Babae G-R. Effect of gamma irradiation on lymphocyte proliferation and IL-8 production by lymphocytes isolated from platelet concentrates. Archives of medical research. 2008;39(6):590-3.

Reviewer 1: Please clarify the statement: (non-proliferative but potentially antigen-presenting cells)

Authors: The following explanation was added to the manuscript, along with relevant references: 

“Irradiation prevents the proliferation of stimulator cells by inhibiting DNA replication but has no effect on transcription, preserving their ability to induce allogeneic T cells proliferation”.

Reviewer 1: Image (1and 4) quality need to improve

Authors: In response to the reviewer's comment, the resolution of Figure 1 was increased. As a result, a new "S1 raw images" file was provided as a Supporting Information file. This Figure was imaged by ChemiDoc™ MP System (Bio-Rad, USA) and was analyzed with Image Lab™ Software. Figure 4's quality was also improved.

Reviewer 1: Histogram plots in Figure 3A quality is poor, Please replace the images with higher resolution images.

Authors: A higher resolution image was used to replace Figure 3. 

Reviewer 1: The caption need clarification” (A) Histogram plots show M1 (divided) and M2 (highly 270 divided) populations of untreated (Light gray) and PHA conditions (dark gray)”

Authors: The caption was edited; “Histogram plots depict examples of two populations used in this study for MLR analysis: M1 (divided) and M2 (highly divided) populations of untreated (Light gray) and PHA conditions (dark gray). M2 is a subset of the M1 population that contains cells with a high division rate.”

Reviewer 1: The results of T cell Proliferation (The M1 and M2 population) should be explained in detail in discussion.

Authors: In response to the respected reviewer's comment, additional explanations about T cell Proliferation (The M1 and M2 population) were added to Result and Discussion sections:

The Result Part: “Indeed, the M2 is a subset of the M1 population that contains cells with a high division rate. The results of the statistical significance (p-value of <0.05) were the same for both populations (Fig 3B).”

The Discussion Part:” In order to analyze the MLR results, M1 (divided) and M2 (highly divided) populations were considered. The statistically significant results (p-value of 0.05) of the two populations were the same, suggesting that the M2 population is primarily responsible for the proliferation results in this study.”

---

## [Decision Letter · Decision Letter 2]

26 Sep 2022

Triggering of lymphocytes by CD28, 4-1BB, and PD-1 checkpoints to enhance the immune response capacities

PONE-D-22-10263R2

Dear Dr. Ramezani,

We’re pleased to inform you that your manuscript has been judged scientifically suitable for publication and will be formally accepted for publication once it meets all outstanding technical requirements.

Kind regards,

Suzie Chen

Academic Editor

PLOS ONE

Additional Editor Comments (optional):

Reviewers' comments:

Reviewer's Responses to Questions

**Comments to the Author**

1. If the authors have adequately addressed your comments raised in a previous round of review and you feel that this manuscript is now acceptable for publication, you may indicate that here to bypass the “Comments to the Author” section, enter your conflict of interest statement in the “Confidential to Editor” section, and submit your "Accept" recommendation.

Reviewer #1: (No Response)

2. Is the manuscript technically sound, and do the data support the conclusions?

Reviewer #1: Yes

3. Has the statistical analysis been performed appropriately and rigorously? 

Reviewer #1: Yes

4. Have the authors made all data underlying the findings in their manuscript fully available?

Reviewer #1: Yes

5. Is the manuscript presented in an intelligible fashion and written in standard English?

Reviewer #1: Yes

6. Review Comments to the Author

Reviewer #1: The authors have responded to some comments made in the previous round of review and have improved the manuscript. accept

7. PLOS authors have the option to publish the peer review history of their article (what does this mean?). If published, this will include your full peer review and any attached files.

Reviewer #1: **Yes: **Amir Ghaemi

---

## [Editor Report · Acceptance letter]

30 Sep 2022

PONE-D-22-10263R2 

Triggering of lymphocytes by CD28, 4-1BB, and PD-1 checkpoints to enhance the immune response capacities 

Dear Dr. Ramezani:

I'm pleased to inform you that your manuscript has been deemed suitable for publication in PLOS ONE. Congratulations! Your manuscript is now with our production department. 

Kind regards, 

on behalf of

Dr. Suzie Chen 

Academic Editor

PLOS ONE